# Hot-Electron Microwave Noise and Energy Relaxation in (Be)MgZnO/ZnO Heterostructures

Emilis Šermukšnis [1],*, Artūr Šimukovič [1], Vitaliy Avrutin [2], Natalia Izyumskaya [2], Ümit Özgür [2] and Hadis Morkoç [2]

[1] Fluctuation Research Laboratory, Center for Physical Sciences and Technology, Saulėtekio Ave. 3, LT-10257 Vilnius, Lithuania; arturas.simukovic@ftmc.lt

[2] Department of Electrical and Computer Engineering, Virginia Commonwealth University, Richmond, VA 23284, USA; vavrutin@vcu.edu (V.A.); nizioumskaia@vcu.edu (N.I.); uozgur@vcu.edu (Ü.Ö.); hmorkoc@vcu.edu (H.M.)

* Correspondence: emilis.sermuksnis@ftmc.lt

**Abstract:** Pulsed hot-electron microwave noise measurements of the (Be)MgZnO/ZnO heterostructures are presented in this work. The heterostructures of different barrier thicknesses and different bulk electron densities in ZnO layer are compared. Capacitance–voltage (C–V) measurements reveal the decrease in the two-dimensional electron gas (2DEG) peak in electron density profile at the Zn-polar BeMgZnO/ZnO interface as the BeMgZnO barrier layer thickness decreases. For thin-barrier heterostructures, the peak disappears and only the bulk electron density is resolved in C–V measurements. The excess noise temperature at ∼10 GHz in thick-barrier heterostructures is noticeably higher (∼10 times) compared to thin-barrier heterostructures, which is attributed to the strong noise source in the contacts of the former. In the case of thin-barrier heterostructures, at electric fields above ∼10 kV/cm and electron density $\gtrsim 1 \times 10^{17}$ cm$^{-3}$, strong noise source is resolved, which was also observed earlier in the Ga-doped ZnO films due to the formation of self-supporting high-field domains. However, for the low electron densities ($\lesssim 6 \times 10^{16}$ cm$^{-3}$), the aforementioned noise source is not observed, which suggests the importance of a deep ZnO/GaN interface with 2DEG for power dissipation. The hot-electron temperature dependence on the dissipated power of those low-electron-density heterostructures is similar to that of O-polar ZnO/MgZnO. The estimated electron energy relaxation time in ZnO/MgZnO is ∼0.45 ps ± 0.05 ps at dissipated electrical power per electron of ∼0.1 nW/el and approaches ∼0.1 ps as the dissipated power is increased above ∼10 nW/el.

**Keywords:** zinc oxide; BeMgZnO/ZnO; ZnO/GaN; TLM; 2DEG; hot electrons; hot phonons; microwave noise



## 1. Introduction

Zinc-oxide (ZnO) as a wide-bandgap semiconductor with high breakdown field (3 MV/cm) and high electron peak velocity ($\gtrsim 3 \times 10^7$ cm/s [1]) is interesting for electronics applications. Doped ZnO films have a base of thin-film transistors (TFTs) [2,3] and transparent conductive electrodes [4]. ZnO-based thin films and TFTs also find interesting applications as sensors [5–8]. Spontaneous and piezoelectric polarization in ZnO creates high-density two-dimensional-electron gas (2DEG) in ZnO-based heterojunctions that are used for heterostructure field effect transistors (HFETs) [9]. A higher electron mobility in the channel of a transistor often assumes a faster operation at a higher power if combined with a high electron density. Because of strong scattering by ionized impurities and defects, the electron mobility is low in the case of doped thin films and a viable alternative for the doped ZnO is a 2DEG channel formed at a heterojunction. MgZnO/ZnO HFETs exploit the 2DEG channel [10–13]. A new trend is based on quaternary BeMgZnO/ZnO heterostructures: in particular, the incorporation of Be is advantageous for achieving a higher 2DEG density [14].

A HFET channel operates at a high electric field, and hot-electron effects manifest themselves. The hot-electron mobility is lower at an elevated electron gas temperature dependent

on the supplied electric power, and the hot-electron energy relaxation time, an additional important parameter, comes into play. The convenient method for measuring the electron energy relaxation time in a high-density 2DEG channel is based on the microwave noise technique [15]. The ZnO 3DEG channels demonstrate a non-monotonous dependence of the relaxation time on the electron density [16]. Hot-electron energy relaxation times in ZnO have been obtained from absorption–transmission and reflectance [17–23], photoluminescence [24–30], femtosecond time-resolved two-photon photoemission experiments (TR-2PPE) [31] and electric-field-dependent photoluminescence measurements for lattice temperatures in the range from 15 K to 90 K [32]. The reported electron energy relaxation spans from 30 fs to 1.75 ps.

In this work, we investigate the hot-electron microwave ($\sim$10 GHz) noise in the gateless (Be)MgZnO/ZnO channels supplied with two ohmic electrodes. The study is conducted for the Zn-polar grown BeMgZnO/ZnO heterostructures with different barrier layer thicknesses and different electron densities in the ZnO layer as well as for the O-polar grown ZnO/MgZnO heterostructure. The electron density profile is measured by a capacitance–voltage (C–V) technique applied to Schottky diodes fabricated on the same structures. The hot-electron noise temperature at $\sim$10 GHz was found to be much higher for thick-barrier layer heterostructures. In this case, noise primarily originates from the contact resistance. In the case of the thin barrier, the hot-electron noise characteristics are similar to those of Ga-doped bulk ZnO films described earlier. For the special case of low electron density in ZnO layer of thin-barrier heterostructures, noise acquires properties of 2DEG channels as a result of a deep ZnO/GaN interface. The dependence of the electron energy relaxation time on the dissipated power per electron $P_e$ for ZnO-based 2DEG channels is found in the wide range of $P_e$. The energy relaxation depends on $P_e$ and is slower at a lower dissipated power.

## 2. Samples and Measurement Techniques

The Zn-polar BeMgZnO/ZnO/GaN structures were grown using plasma-assisted molecular beam epitaxy (PA-MBE) on (0001)-GaN/sapphire templates. Highly resistant (carbon-compensated) 2.5 µm-thick Ga-polar GaN layers were grown via metal–organic chemical vapor deposition on (0001) sapphire. The polarity of the ZnO buffers was inverted from Zn-polar to O-polar by employing a high VI/II ratio during nucleation [33]. The polarity inversion was followed by annealing at 730 °C for 5 min at a reactor pressure of $10^{-5}$ Torr with the oxygen plasma shutter closed, and then a 130 nm-thick ZnO buffer layer was grown at 670 °C under oxygen-rich conditions (VI/II ratio 6.0). Then, the growth of the quaternary $Be_{0.02}Mg_{0.26}ZnO$ layer under an oxygen-rich environment and a reactor pressure of $1.5 \times 10^{-5}$ Torr began. The oxygen plasma cell served as a source of reactive oxygen. The metals were evaporated from the K-cells. The detailed growth procedure can be found here [34].

Several samples with different BeMgZnO barrier thicknesses were prepared. Hall measurements in the van der Pauw configuration provided mobility of $169 - 226$ cm$^2$/Vs and sheet electron density of $(8.7 - 9.5) \times 10^{12}$ cm$^{-2}$ in samples with barrier thickness of $\gtrsim$20 nm. The corresponding values for $\lesssim$20 nm were $90 - 117$ cm$^2$/Vs and $(6 - 13) \times 10^{12}$ cm$^{-2}$.

Also, an O-polar ZnO/MgZnO/ZnO/MgO structure was grown on c-plane sapphire substrate by the PA-MBE. A 2 nm-thick MgO layer was deposited on the c-plane sapphire, the deposition was continued with a 35 nm-thick ZnO high-temperature buffer layer, a 100 nm-thick $Zn_{0.54}Mg_{0.46}O$ barrier layer, and a 35 nm-thick ZnO channel layer. The Hall mobility and density were 156 cm$^2$/Vs and $6 \times 10^{12}$ cm$^{-2}$.

Cross sections of the samples are presented in Figure 1a. In Zn-polar structures, 2DEG forms near the bottom surface of BeMgZnO barrier, while in O-polar structures it forms near upper surface of MgZnO barrier (2DEG is indicated by red dashed lines). The formation of 2DEG is mainly determined by the discontinuity of polarization at the heterointerface. In the case of Zn-polar growth, the discontinuity of polarization creates a positively charged surface at the bottom surface of the BeMgZnO barrier layer, resulting in the formation of 2DEG in the ZnO layer adjacent to the heterointerface. In the case of O-polar growth, the positively charged surface is at the top of the MgZnO barrier layer, giving rise to the 2DEG

channel in ZnO just above the heterointerface. Therefore, the differences in the position of 2DEG are due to differences in growth polarity. A more detailed discussion of the 2DEG formation in the Zn- and O-polar ZnO/(Be)MgZnO interfaces can be found in the work of Ding et al. [35]. The same principle works for the GaN-based systems of Ga-polar and N-polar growth [36,37].

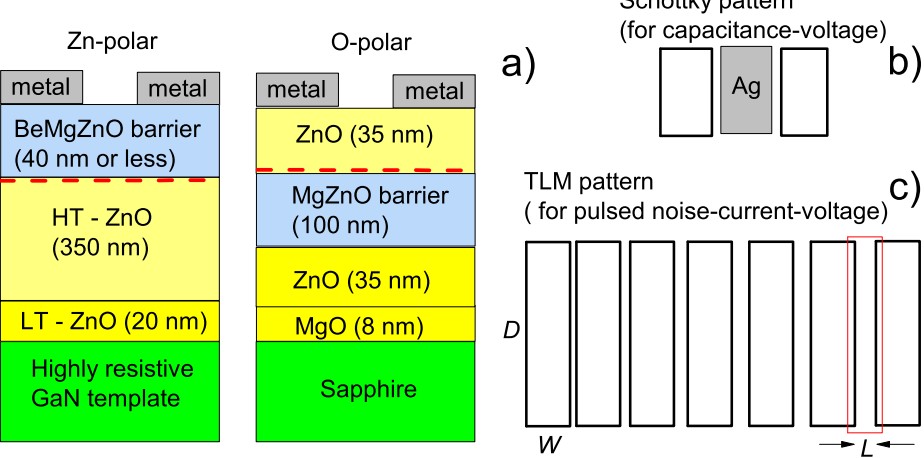

**Figure 1.** Schematic structure of the Zn-polar BeMgZnO/ZnO and O-polar ZnO/MgZnO heterostructures with metal contacts (**a**), pattern for C–V measurements (the middle electrode is Schottky, and the two side electrodes are ohmic) (**b**) and a transmission line model pattern for pulsed microwave noise measurements (**c**). The red dashed solid lines indicate the presence of 2DEG at the interfaces. The length and the width of rectangular electrodes are $D$ and $W$, respectively. The gap $L$ between two adjacent electrodes increases from 4 μm to 27 μm.

C–V measurements are conducted on planar-rectangular Schottky diode patterns, consisting of three electrodes where the middle one is a (Ag) Schottky contact, and two electrodes on the sides are ohmic contacts (Figure 1b). Dimensions are $W = 80$ μm and $D = 140$ μm. Transmission line model (TLM) patterns are used for hot-electron microwave noise measurements (Figure 1c). TLM patterns consist of rectangular electrodes separated by gaps of increasing length, ranging from 4 μm to 27 μm. The width $W$ and the length $D$ of rectangular electrodes are $W = 80$ μm and $D = 300$ μm. Smaller electrodes are used, too, with dimensions of $W = 55$ μm and $D = 170$ μm.

Hot-electron microwave noise is measured in TLM structures (the set of gateless channels) by placing a two-pin microwave probe (0–40 GHz band) on the adjacent rectangular electrodes separated by the gap $L$ (the length of the semiconductor channel). The microwave probe can be contacted on different pairs of electrodes of the TLM to select any $L$ channel for measurements. A pulsed voltage generator is connected to the sample; 30 ns, 100 ns or 3 μs length and low-duty-cycle ($\sim 10^{-5}$) voltage pulses bias the sample and create pulsed electric field $E$ in the channel up to $\sim 100$ kV/cm. Pulsed measurements at high fields help to mitigate the Joule heating of the sample. Electric field $E$ is estimated simply from applied pulsed voltage on the channel of length $L$.

The sample heated by the electric field emits microwave noise. The noise signal is amplified by a low-noise amplifier (LNA) and is detected by a gated radiometer system (the gate length is set to be approximately twice as short as the voltage pulse length) which measures an average noise power. After the calibration procedure (by using a standard noise source), noise power is converted to excess noise temperature $\triangle T_\mathrm{n} = T_\mathrm{n} - T_0$, where $T_\mathrm{n}$ is the noise temperature and $T_0$ is the room temperature (i.e. sample temperature at equilibrium). The pulsed voltage and current in the sample are monitored at the oscilloscope. Both (pulsed) waveforms are acquired by inserting coaxial voltage and current probes between the pulse generator and the sample. More details on the microwave noise measurement setup can be found elsewhere [15,38]. The C–V characteristics are measured on

rectangular Schottky patterns (Figure 1b) with the use of an LCR meter operating at 2 MHz stimulus frequency. The apparent electron density dependence on distance (i.e., electron density profile) is then calculated [37]. Noise and C–V measurements are performed at RT.

## 3. Results

Electron density profile measurements for heterostructures with different barrier thicknesses are presented in Figure 2. The peaks show the formation of 2DEG near Be-MgZnO/ZnO interfaces and the position of the peaks reveals the thickness of the BeMgZnO barrier layer. The peaks slowly diminish as the barrier thickness is decreased from approximately ~40 nm to ≲20 nm. So, the electron gas confinement becomes poor at barrier thickness of ≲20 nm. A similar tendency was also observed in GaN-based heterostructures [39,40]. A further decrease in barrier thickness results in the disappearance of the 2DEG peak (Figure 3). The electron density profiles of thin-barrier heterostructures of different bulk electron densities in ZnO layer are presented in Figure 3. There is some small variation in the barrier thickness per sample (see the group of sample A2 in Figure 2 as an example). The same variation in sample B1 gives a weak onset of the 2DEG peak, while for slightly larger barrier thickness, the peak almost disappears (Figure 3). The variation of bulk electron density in the ZnO layer also can have an affect on the peak formation.

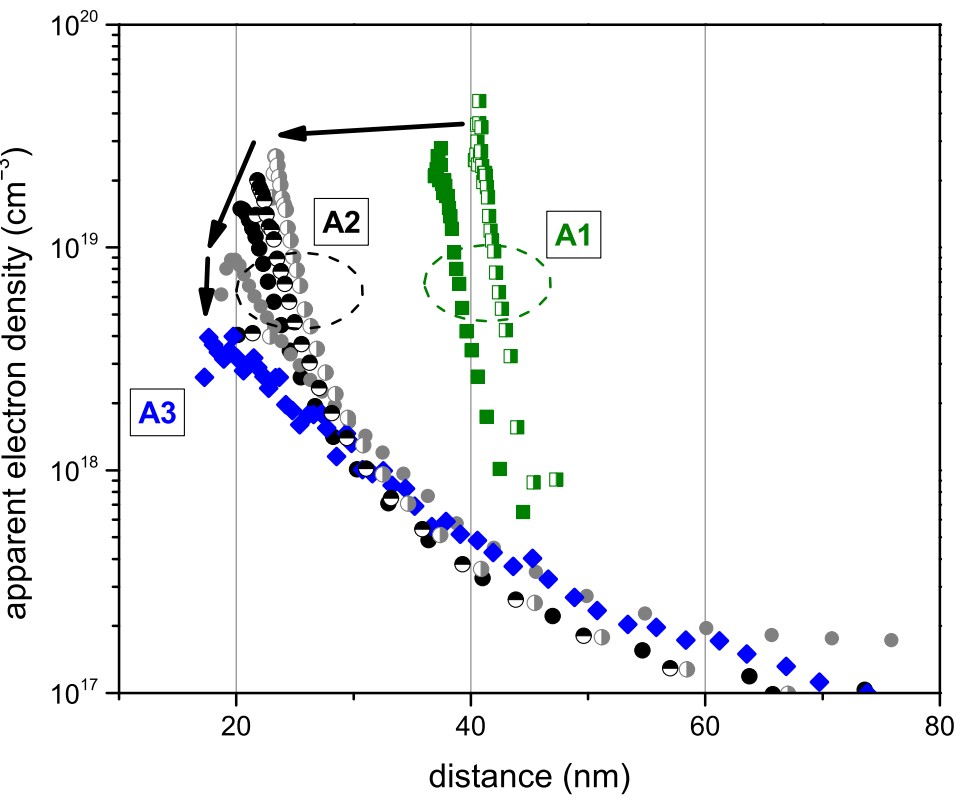

**Figure 2.** The dependence of apparent electron density on the distance for thick-barrier layer BeMgZnO/ZnO heterostructures. Squares, circles and diamonds stand for samples A1, A2 and A3, respectively.

Microwave noise characteristics for thick-barrier (samples A1, A2 and A3) and thin-barrier heterostructures (samples B1, B2, B3 and B4) are compared in Figure 4. At lower fields ($E < 1$ kV/cm and $E < 10$ kV/cm for thick- and thin-barrier channels, respectively) the trend of excess noise temperature $\triangle T_n$ follows $\triangle T_n \propto E^2$ (see lines). In the case of thick barriers, the excess noise temperature is nearly 10 times higher for the same electric field, on average. The excess noise temperature approaches almost ~10,000 K at higher field, which is much higher than the room temperature.

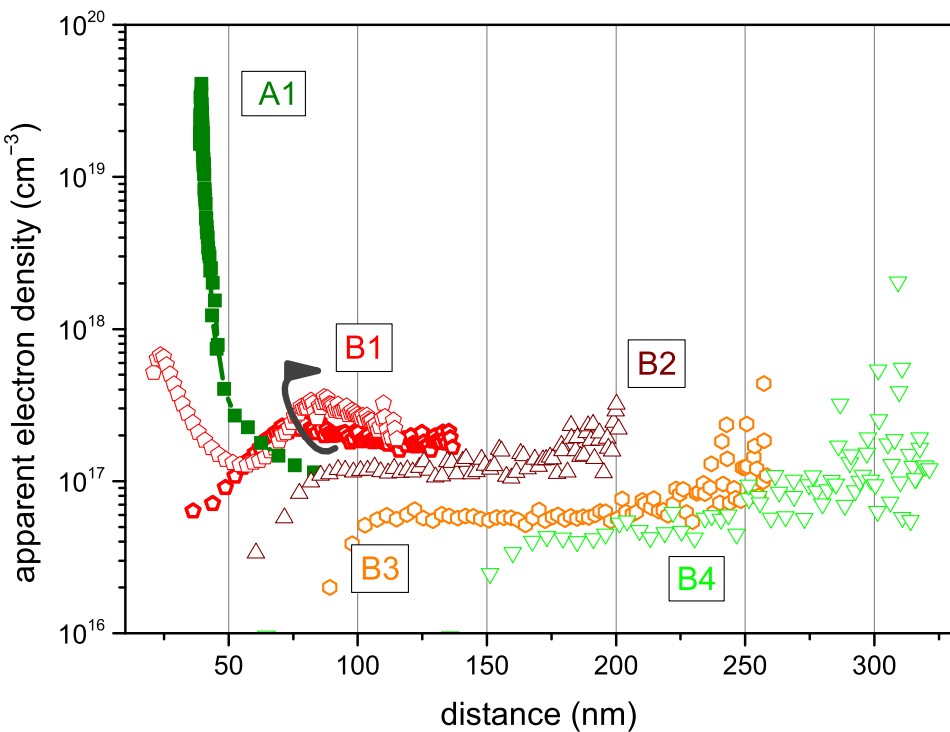

**Figure 3.** The dependence of apparent electron density on the distance for thin-barrier layer Be-MgZnO/ZnO heterostructures of different bulk electron density in ZnO layer. Pentagons, up-triangles, hexagons and down-triangles stand for thin-barrier samples B1, B2, B3 and B4, respectively. Sample A1 with a thick-barrier layer is shown for reference.

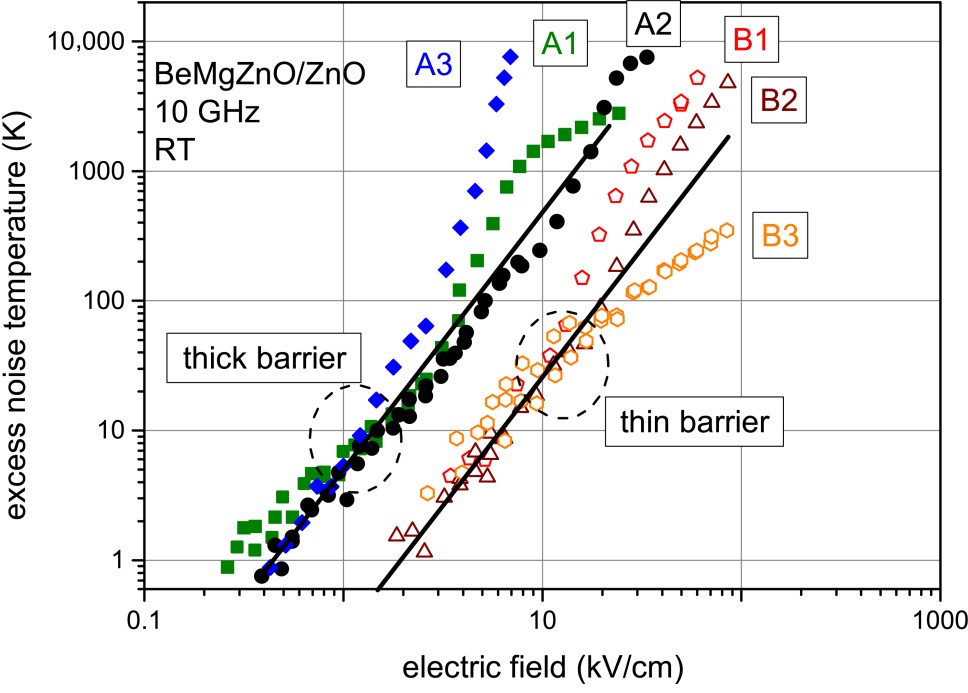

**Figure 4.** The dependence of excess noise temperature on electric field in BeMgZnO/ZnO heterostructures. Closed and open symbols stand for thick-barrier and thin-barrier samples, respectively. Lines are $\triangle T_{\mathrm{n}} \propto E^2$.

## 4. Discussion

The noise data for different-length (*L*) channels of thick-barrier heterostructure (sample A1, ∼40 nm barrier thickness) are shown in Figure 5. The different-length channels are compared in the noise-current representation, which allows us to evaluate the effect of contact resistance [41]. In particular, the excess noise temperature strongly decreases as *L* increases for thick-barrier sample A1. Such a strongly decreasing dependence on *L* is an indication that the noise is mainly a result of the contact resistance [41]. However, in the case of thin-barrier samples, $\triangle T_n$ does not noticeably decrease on *L*. Moreover, the dependence shows the opposite trend in some thin-barrier samples at higher currents (Figure 6). The dependence on *L* is weak, or perhaps even non-existent, for the whole measured current range of sample B3 (Figure 7). Therefore, in the thin-barrier samples, the noise mainly originates from the channels. Thin-barrier heterostructures are further studied because hot-electron noise in this case is a fingerprint of the power dissipation properties of the conductive channels but not of the contacts.

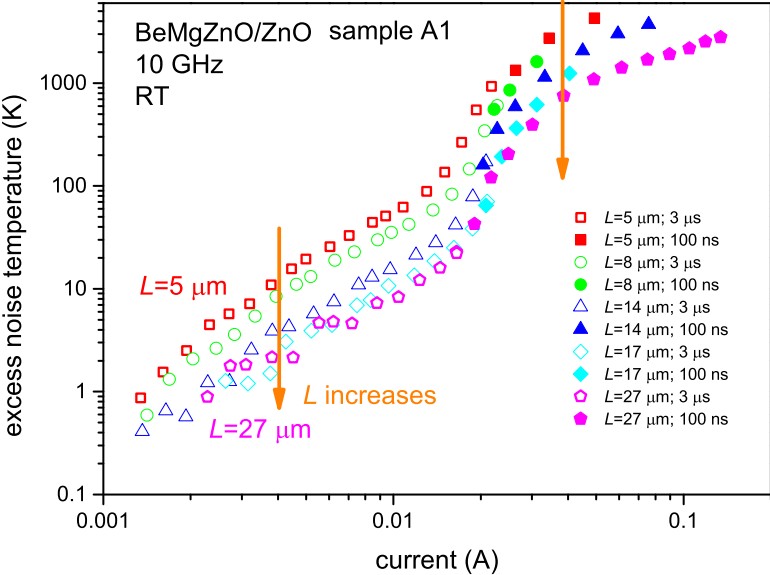

**Figure 5.** The dependence of excess noise temperature on current for different channel lengths (5 μm–27 μm) in thick-barrier BeMgZnO/ZnO sample A1. Pulse length is 3 μs (open symbols) and 100 ns (closed symbols).

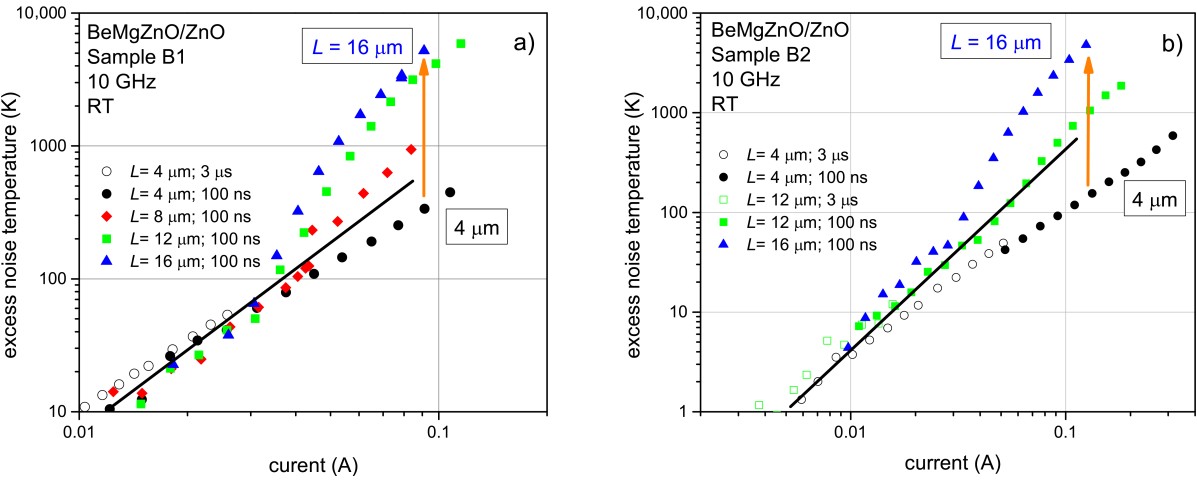

**Figure 6.** The dependence of excess noise temperature on current for different-length channels of thin-barrier BeMgZnO/ZnO samples B1 (**a**) and B2 (**b**). Open and closed symbols stand for 3 μs and 100 ns pulse width, correspondingly. Lines are $\triangle T_n \propto I^2$.

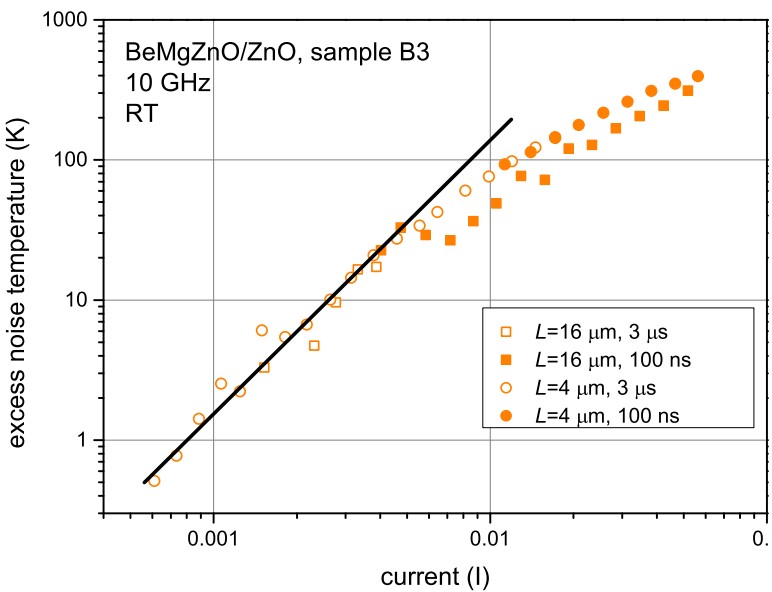

**Figure 7.** The dependence of excess noise temperature on current in BeMgZnO/ZnO sample B3 for different channel lengths. Pulse length is 3 μs (open symbols) and 100 ns (closed symbols).

As was noted, thin-barrier samples B1 ($n_{C-V} \approx 2 \times 10^{17}$ cm$^{-3}$) and B2 ($n_{C-V} \approx 1 \times 10^{17}$ cm$^{-3}$) show opposite dependence on $L$ (see Figure 6, at $I > 35$ mA, for samples B1 and B2) than in the thick-barrier sample A1. The threshold-like onset of noise temperature shows up for all channels with $L > 4$ μm of samples B1 and B2. The noise onset is stronger for the longer channels in which excess noise temperature lifts over the $\triangle T_n \propto I^2$ line, plotted through experimental noise points at low bias, forming bump-like dependence. A similar trend was observed in bulk Ga-doped ZnO films of different 3D electron densities and it was attributed to the high-field domain formation [42]. The formation was resolved for the Hall electron density range of $n_{Hall} \approx 1.4 \times 10^{17}$ cm$^{-3} - 2.4 \times 10^{18}$ cm$^{-3}$, while at a densities above that range, as well as in the case of ZnO-based 2DEG channels, it was not. Therefore, microwave noise in thin-barrier samples B1 and B2 resembles that of bulk films. This also agrees with C–V profiling plots in Figure 3, in which no 2DEG is observed and 3D electrons dominate.

The same can be said about the C–V profiles of even smaller bulk electron density samples B3 ($n_{C-V} \approx 6 \times 10^{16}$ cm$^{-3}$) and B4 ($n_{C-V} \approx 4 \times 10^{16}$ cm$^{-3}$), which also show no signs of 2DEG peaks. However, in those two samples, no threshold-like noise onset is resolved in contrast to the higher-electron-density samples B1 and B2 (compare Figures 6 and 7). In particular, the excess noise temperature dependence (even for 14-μm-long channel) goes below the $\triangle T_n \propto I^2$ line (Figure 7). The same qualitative difference is also seen in Figure 4 (compare the thin-barrier samples).

The absence of the aforementioned noise onset is a property either of ZnO-based 2DEG channels or higher-electron-density ($>2.4 \times 10^{18}$ cm$^{-3}$) bulk-film channels according to [42]. As the electron densities in samples B3 and B4 are both $\ll 2.4 \times 10^{18}$ cm$^{-3}$, we are left only with a 2DEG option (high-field domains do not form) but this contradicts the C–V profiles, which show no 2DEG peak for samples B3 and B4 (see Figure 3).

This contradiction could be reconciled if the presence of the 2DEG gas at the ZnO/GaN interface is considered [43]. In BeMgZnO/ZnO samples, the GaN template on sapphire is used (see Figure 1a). Due to the limited depth of the C–V scan, the ZnO/GaN interface is not reached. According to Figure 1a, the position of the ZnO/GaN interface is >370 nm (i.e., 370 nm + the width of the barrier) from the surface. The C–V scan depth reaches only ~320 nm for sample B4, and even less for other samples, as shown in Figure 3. Therefore, C–V profiling does not prove the absence of a deep 2DEG channel at the ZnO/GaN interface.

To reiterate, thin-barrier heterostructures, depending on the electron density of the ZnO layer (obtained from C–V measurements), show qualitatively different hot-electron noise

characteristics for higher fields. In particular, the absence of threshold-like noise onset in the lower-electron-density channels indicates the presence of the deep 2DEG channel. Though the deep channel is present in all Zn-polar structures discussed here, it manifests itself in noise characteristics only in some of them. This can be understood from the simplistic model of two parallel channels; the 350 nm-thick ZnO channel with 3DEG and the ZnO/GaN channel with 2DEG. In the case of low 3DEG density in the ZnO layer, the resistance of this layer will be high, and, as a result, conductance of 2DEG at the ZnO/GaN interface will dominate.

It is important to note that the dependence of excess noise temperature on dissipated power per electron $\triangle T_n(P_e)$ of sample B3 is close to that of the 2DEG ZnO/MgZnO channel (Figure 8). Dissipated power $P_e = U \times I/N$ is estimated from the current and voltage pulse amplitudes, $I$ and $U$, and a number of electrons $N$ in the conductive channel obtained from the Hall measurements. There is no dependence on pulse length, which means that self-heating (Joule heating) is controlled by using short pulses.

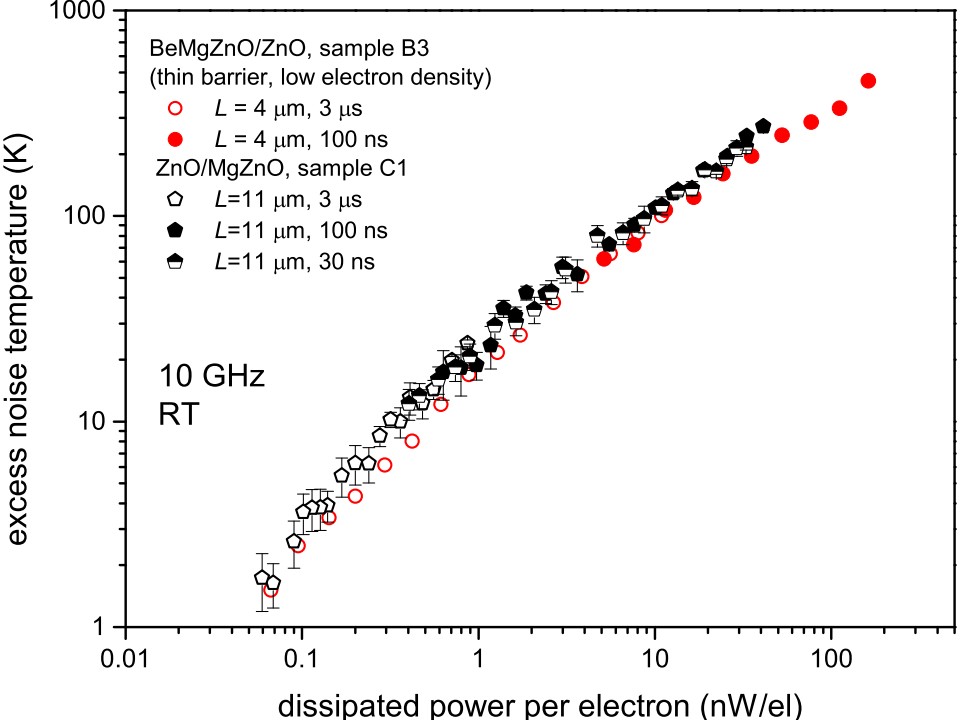

**Figure 8.** The dependence of excess noise temperature on electric field in Zn-polar BeMgZnO/ZnO sample B3 (circles) and O-polar ZnO/MgZnO sample C1 (pentagons). Pulse length is 3 μs (open symbols), 100 ns (closed symbols) and 30 ns (half-full symbols).

Electron energy relaxation time $\tau_e$ can be estimated as [44]:

$$\tau_e = \frac{k_B \triangle T_e}{P_e}, \tag{1}$$

where $k_B$ is the Boltzmann constant and $\triangle T_e$ is the excess electron temperature. The excess electron temperature is $\triangle T_e = T_e - T_0$, where $T_e$ is an electron (gas) temperature. At microwave frequencies ($\sim 10$ GHz), the thermal noise dominates while low-frequency noise sources like flicker noise ($1/f$) or generation-recombination noise are not important. Similarly to the noise thermometry method, hot-electron (thermal) noise temperature can be used as a measure of hot-electron temperature according to approximate relation $T_e \simeq T_n$ [45].

The dependence of electron energy relaxation time on dissipated power per electron is depicted in Figure 9a. At lower dissipated power of $\sim 0.1$ nW/el, the electron energy relaxation time is $\sim 0.4$ ps–0.5 ps for the ZnO/MgZnO channel and slightly lower ($\sim 0.35$ ps)

for thin-barrier BeMgZnO/ZnO/GaN sample B3. At higher dissipated power (and higher excess electron temperature), the relaxation *rate* increases, and $\tau_e$ reaches $\sim 0.1$ ps as $P_e$ exceeds $\sim 10$ nW/el. The obtained values $\sim 0.1$ ps–0.5 ps are comparable with those estimated for Ga-doped bulk ZnO films of electron densities $< 2 \times 10^{19}$ cm$^{-3}$ [16].

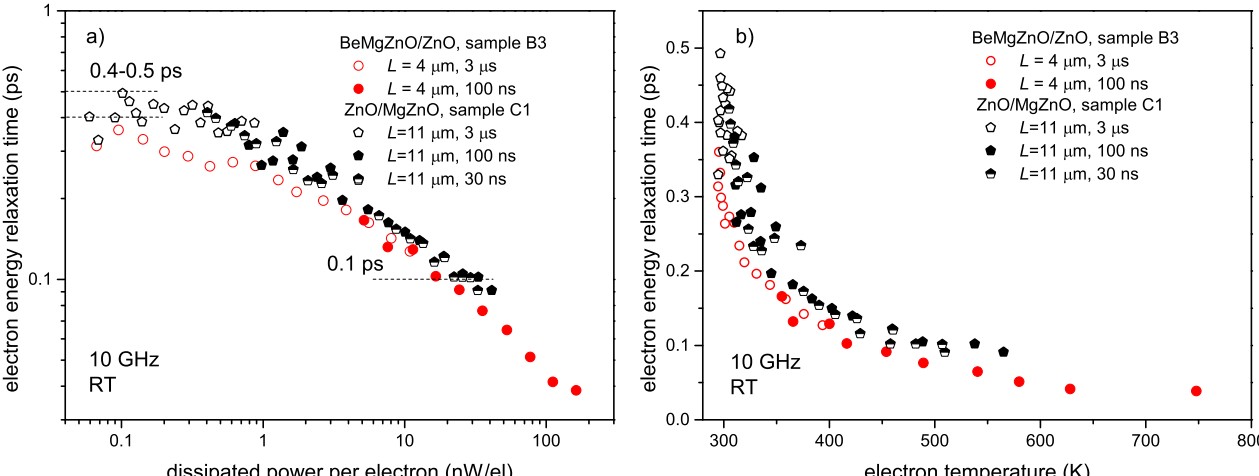

**Figure 9.** The dependence of electron energy relaxation time on dissipated power per electron (**a**) and electron temperature (**b**). Zn-polar BeMgZnO/ZnO sample B3 and O-polar ZnO/MgZnO sample C1 correspond to circles and pentagons, respectively. The pulse length is 3 μs (open symbols), 100 ns (closed symbols) and 30 ns (half-full symbols).

The power dissipation in 2DEG channels is slowed by accumulation of inequilibrium (hot) longitudinal optical (LO) phonons [46,47]. The decrease of energy relaxation time in *GaN*-based 2DEG channels was related to phonon–plasmon resonance (and associated decrease in the LO-phonon lifetime) [48]. On the other hand, as quasi-two-dimensional electron gas at the barrier/ZnO interface heats, the occupancy of higher-energy subbands can increase, resulting in some additional spread of the electron-density profile into the ZnO layer and a decrease of peak electron density (for 2DEG-based channel, see [48]). The excess hot-electron temperature reaches 200–400 K (see Figure 8). The decrease in peak electron density (per unit volume) can mitigate the effect of hot LO phonons on the energy relaxation (even for a constant LO-phonon lifetime) which depends on electron density [45,49]. The model of diffusion of (hot) LO phonons in the Brillouin zone might also be important [50].

The dependence of $\tau_e$ on electron temperature $T_e = \triangle T_e + T_0$ is plotted in the Figure 9b. The fast decrease in $\tau_e$ at low electron temperature range ($T_e < 500$ K) as well as a similar value of $\sim 0.4$ ps–0.5 ps were also obtained for the GaN-based 2DEG channel by a model when hot LO phonons and Debye screening were included [51]. A slightly lower $\tau_e$ value was estimated in a case of dynamic screening. In GaN-based channels of higher 2DEG density, even $\sim 5$ ps was measured [38]. However, the aforementioned models and experiments were not applied to the ZnO-based heterostructures.

In conclusion, hot-electron noise characteristics of Zn-polar grown BeMgZnO/ZnO and O-polar grown ZnO/MgZnO heterostructures were investigated. Noise temperature characteristics at 10 GHz were found to be different depending on the BeMgZnO barrier thickness available from capacitance–voltage electron density profile measurements. The noise temperature was considerably higher for the thick-barrier ($\gtrsim 20$ nm) heterostructures showing a 2DEG peak in C–V profiling compared to the thin-barrier ($\lesssim 20$ nm) heterostructures in which no 2DEG peak was resolved. While the noise of Zn-polar heterostructures with thick barriers was mostly dominated by contacts, in the case of thin barriers it was dominated by the conductive channel. In latter case, the noise characteristics were similar to those of Ga-doped bulk films, in which a high-field self-supporting domain formation was reported previously. In the special case of low electron density ($n_{C-V} \lesssim 6 \times 10^{16}$ cm$^{-3}$)

in ZnO layers of thin-barrier structures, the deep ZnO/GaN channel dominated noise characteristics and power dissipation; the hot-electron temperature and energy relaxation time in this case were similar to those of the O-polar MgZnO/ZnO 2DEG channel. At low dissipated power ($\sim$0.1 nW/el) the energy relaxation was slow ($\sim$0.45 ps $\pm$ 0.05 ps), while at a higher dissipated power ($\gtrsim$10 nW/el), the electron energy relaxation approached $\sim$0.1 ps.

**Author Contributions:** Conceptualization, E.Š.; methodology, E.Š and V.A.; software, E.Š.; validation, E.Š. and V.A.; formal analysis, E.Š.; investigation, E.Š., A.Š., V.A. and N.I.; resources, H.M. and Ü.Ö.; writing—original draft preparation, E.Š.; writing—review and editing, E.Š., V.A. and N.I.; visualization, E.Š. and A.Š.; supervision, H.M.; project administration, V.A. and Ü.Ö.; All authors have read and agreed to the published version of the manuscript.

**Funding:** This research received no external funding.

**Data Availability Statement:** The data presented in this study are available on request from the corresponding author. The data are not publicly available due to privacy concerns.

**Acknowledgments:** The authors thank former member of the VCU group K. Ding who participated in growth and fabrication of the ZnO samples included in this research. The authors also thank former member of CPST group, J. Liberis who participated in the experimental part of the research.

**Conflicts of Interest:** The authors declare no conflicts of interest.

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
