# Peer review of "Hot-Electron Microwave Noise and Energy Relaxation in (Be)MgZnO/ZnO Heterostructures"

_crystals, doi:10.3390/cryst14010075_

Round 1

Reviewer 1 Report

Comments and Suggestions for Authors

The presented manuscipt concerns an important topic of 2DEG properties in HFETs and provides interesting insights into the examined physics. The text is very well written, however, somewhat heavy and difficult to follow because of dense information contained; a text lightening would make it easier and more comprehensible. Some comments for Authors:

1. Laboratory names of samples like '1208', '1187N2', '1187O2' are not informative for a reader and are advised to be renamed to meaningful ones or to simple 'ABC' or '123'.

2. In lines 171-177 Authors discuss controversal results mentioning the presence of a 2DEG at ZnO/GaN interface. They noticed that it lies deep in the structure (>350 nm), therefore beyond the reach of C-V scan. What is the distance of C-V scan that can be reached? If the mentioned ZnO/GaN interface does not affect the measurements results, why taking it into consideration at all?

3. In addition to the comment 2.: the other structure, O-polar ZnO/MgZnO, features two ZnO/MgZnO interfaces, because the MgZnO layer is sandwitched between two 35 nm ZnO layers. Why the 2DEG is marked in the schematics only at the upper interface? The MgZnO layer is 100 nm - is it reachable for a C-V scan?

An outlook for the presented study in the conclusions would be desirable.

Reviewer 2 Report

Comments and Suggestions for Authors

The paper presents a study of (Be)MgZnO/ZnO heterostructures predicted to be used for field effect transistors. They are characterized by capacitance voltage (C-V) and hot electron microwave noise (HEMN) measurements. The C-V experimental data reveal a peak in the electron density profile near the (Be)MgZnO/ZnO interface. This peak vanishes for thin (Be)MgZnO layers. Besides, the HEMN is drastically increased for thick (Be)MgZnO layers. Therefore, in the case of thick (Be)MgZnO films, the HEMN enhancement is attributed to the electrons close to the (Be)MgZnO/ZnO interface. The experimental data and interpretations are convincing. The article is well written and can be published as is.
